# Influence of the Pedagogical Model and Experience on the Internal and External Task Load in School Basketball

**DOI:** 10.3390/ijerph182211854

**Published:** 2021-11-12

**Authors:** María G. Gamero, Juan M. García-Ceberino, Sergio J. Ibáñez, Sebastián Feu

**Affiliations:** 1Optimisation of Training and Sports Performance Research Group (GOERD), University of Extremadura, 10003 Cáceres, Spain; mgamerob@alumnos.unex.es (M.G.G.); juanmanuel.garcia.ceberino@ui1.es (J.M.G.-C.); sibanez@unex.es (S.J.I.); 2Faculty of Education, University of Extremadura, 06006 Badajoz, Spain; 3Faculty of Sports Science, University of Extremadura, 10003 Cáceres, Spain

**Keywords:** physical–physiological demand, methodology, physical education, basketball, inertial device

## Abstract

The methodology used by the teacher in Physical Education sessions conditions the physical fitness of the students, since the design of the tasks determines the physical and physiological demands to which they are exposed. This study aimed to quantify and compare, according to the teaching methodology and students’ previous experience, the external (eTL) and internal (iTL) load resulting from the application of three intervention programmes that follow different teaching methodologies to teach school basketball: the Tactical Games Approach (TGA), Direct Instruction (DI) and Service Teacher’s Basketball Unit (STBU). The Ratings of Perceived Exertion (RPEs) recorded in the assessments (pre-test/post-test) were also studied. A total of 49 students, aged 11 to 12 and divided into three class groups, from the sixth grade of primary education at a state school in Spain, participated in the study. Teaching–learning programs were randomly assigned to student groups. All the sessions were monitored with inertial devices that made it possible to record physical activity and convert the information into kinematic parameters. The results indicated that during the sessions, the students who followed the TGA method recorded higher values of eTL (player load; DI = 4.92, TGA = 6.95, STBU = 2.99) and iTL (mean heart rate; DI = 142.94, TGA = 157.12, STBU = 143.98). In addition, during the evaluation tests, they presented heart rate levels similar to those obtained by the students in the other programmes. However, they spent more time doing high-intensity activity, working longer in the running (DI = 3.42, TGA = 11.26, STBU = 8.32) and sprinting speed ranges (DI = 0.00, TGA = 0.12, STBU = 0.11), and presenting better physical fitness. During the assessments, students with no prior basketball experience showed higher levels of top speed; experienced students had higher levels of heart rate. The TGA method favours the physical condition and health of primary education students, which is why this method is recommended when planning Physical Education sessions.

## 1. Introduction

The decrease in the practice of physical activity and the rise in the use of new technologies have changed young people’s leisure habits and increasing, in the last decade, concern about their physical and emotional health [1]. This concern has increased after analysing the impact of the negative effects of the current COVID-19 pandemic on young people, as it has promoted sedentary behaviours and emotional imbalance [2]. The World Obesity Federation (WOF) esteem that 254 million children and adolescents will be obese by 2030, becoming the main non-transmissible disease and constituting a serious problem for public health [3]. Thus, a minimum of 60 min of physical activity per day is recommended for school age children; however, not only is the duration of the activity important but also the intensity at which it is practised [4]. There is concern about the short time devoted to physical activity classified as vigorous during Physical Education classes, as it only occupies about 10 to 15% of class time, or approximately five minutes [5]. In this context, Physical Education has been considered to be a subject of great interest due to its potential for developing healthy activities and habits, constituting a fundamental pillar in the integral education of students due to the values and attitudes that are implicitly worked on. Regular practice of physical activity is associated with better physical and mental health; thus, Physical Education is a fundamental pillar in children’s development. As school is where most children practise their physical activity, the planning and methodology used by the teacher in classes is very important for optimising the students’ physical fitness [6]. Slingerland et al. [7] state that physical activity levels can be increased using contact sports like basketball, as its practice involves many low- and high-intensity actions. It also makes it possible for several players to participate at the same time and in the same place through cooperation and opposition [8], favouring personal relations and motor development. Thus, these contents are an interesting proposal to be included in teachers’ programming, as one of the main objectives of this subject is the development of the students’ physical aptitudes [9].

Regarding methodological concepts, it has been shown that some teaching approaches obtain better results in sports learning than others [10]. The Physical Education teacher relies on two basic approaches: the Teacher Centred Approach (TCA), based on traditional methodology, and the Student Centred Approach (SCA), which arose as a more flexible alternative [11]. Within the TCA, the Direct Instruction method is the most typical, while in SCA, the Tactical Games Approach stands out [12]. In the case of the TCA approach, the teacher proposes the initial tasks to develop technical competence using individual practice in repetitive exercises. These tasks generate little motor and cognitive implication in the students [11]. Later on, when the teacher considers that the students have mastered the technique, there is development towards more complex game situations. In this model, the teacher uses prescriptive feedback to correct mistakes [11]. In contrast, the SCA approach proposes tasks that represent tactical problems, thus encouraging students’ decision making [11], based on game situations as contextualised and motivating as possible [13]. The tasks are presented in the form of small-sided games (SSGs), and the teacher uses interrogative feedback so that the students develop decision-making abilities and improve their own tactical awareness. This approach produces greater involvement and participation on the part of the students, to facilitate understanding of the game and improve tactical decision making [14]. Thus the approach used by the teacher conditions the students’ physical aptitude, as it is associated with a differentiated response in the supported load [15,16]. The SCA approach provokes higher heart rates, improving the student’s cardiorespiratory aptitude [17]. It also makes it possible for physical aptitude levels to improve, as well as developing the students’ motor abilities, and is the most effective with the use of invasion sports [18]. It is therefore recommended for designing basketball tasks in Physical Education, in order to provide the students with high-intensity interventions, where at least 50% of the time is spent doing moderate to vigorous physical activity (MVPA) [18]. In spite of the benefits that are attained with the SCA, the TCA is still the one most commonly used, causing low levels of physical activity in the students due to the limited time for motor practice [19]. In the study carried out by Casey [20], the need for teachers to update their practices is concluded after experiencing real differences in the learning of their students and in their effectiveness as teachers. However, in order to achieve this change in approach, continuous training of both teachers and teacher trainers is necessary [21].

The physical–physiological demands supported by the students during a task can be measured with (i) an analysis of the external load (eTL) according to movements in time; and (ii) an analysis of the internal load (iTL) using heart rate (HR) or subjective scales [22]. Buchheit et al. [23] define eTL as the mechanical and locomotor stress caused by an activity (imposed stimulus), differentiating it into neuromuscular and kinematic loads. ITL represents the biomechanical, physical and physiological responses of the players to the stimuli of the activity or competition, which are different for each individual [24]. There are different instruments for measuring load in classes/training sessions: accelerometers and inertial devices can be used to objectively quantify eTL [24]. However, these systems are not usually affordable in school contexts due to their high cost. For this reason, Ibáñez et al. [25] propose an Integral Analysis System of Training Tasks, (SIATE), which entails no cost and can be adapted to all contexts. Several studies have used this system to analyse the subjective loads of teaching tasks [26,27]. For objective iTL measurements, heart rate monitors are used, and one of the most commonly used economic methods is the scale of rated perceived exertion (RPE) [28]. The scientific literature concludes that there is a strong correlation between HR loads and those obtained from RPE in contact sports [29]. There is also a relation between the quantification of subjective and objective eTL and iTL (HR) [30,31]. Thus, Physical Education teachers can reliably monitor the load provoked by their planning, optimising the teaching–learning process. 

Several studies analyse the differences that exist among teaching–learning approaches, mainly focussing on pedagogical and psychological variables [32,33,34]. Differences in the understanding of the sport have also been studied by analysing students’ declarative and procedural knowledge [35,36]. However, few studies have analysed the different responses that are provoked by physical and physiological demands in the school context [6,37]. These investigations use teacher-researchers to apply different methods; however, this study analyses the physical and physiological demands provoked by an intervention programme designed by a Physical Education teacher, a fundamental figure in the teaching-learning process. The aims of this study were (i) to quantify the physical-physiological demands that students experience (eTL, iTL and RPE) during the implementation of three different intervention programs for the teaching of school basketball; (ii) compare this quantification, depending on the students’ previous experience with basketball. The hypotheses of the study are (i) the students who have received the teaching program based on the TGA method will present higher levels of eTL and iTL; (ii) students with previous basketball experience will obtain higher levels of eTL and lower levels of iTL and RPE.

## 2. Materials and Methods

### 2.1. Design

The present study uses a manipulative strategy of a quasi-experimental and longitudinal type [38]. A pre-test/post-test design was used to determine the differences in the physical and physiological demands recorded by the students after the implementation of three intervention programmes based on different methodologies, for teaching school basketball. 

### 2.2. Sample 

A total of 55 students of 11 and 12 years of age from the sixth grade of primary education participated in the study. They were divided into three heterogeneous and mixed groups (6A, 6B and 6C). The investigation was carried out in a state school in the central-western region of Spain. The teaching–learning programmes were randomly assigned to the groups. The students from 6A (*n* = 18) participated in the Direct Instruction (DI) intervention, the students from 6B (*n* = 19) participated in the Tactical Game Approach (TGA), and the students from 6C (*n* = 18) participated on the Service Teacher’s basketball Unit (STBU), designed and imparted by the Physical Education teacher without an explicit definition of the teaching model used. To be part of the study, all the students had to have participated in at least 80% of the practical interventions (sessions and pre-test/post-test assessments). Finally, after 6 subjects dropped out due to experimental mortality, the sample comprised 49 students. The students had not had prior contact with the invasion sport of basketball in their Physical Education classes in previous years. However, a high percentage of students did practise basketball as training/activity out of school for two hours a week; 44% from the DI intervention, 52.6% from the TGA intervention and 27.8% from the STBU programme. Figure 1 shows a flow chart on sample size.

### 2.3. Variables

Two independent variables were determined in the study (i) the teaching–learning methodology used in the different intervention programmes and (ii) the prior experience of the students in basketball.

Three intervention programmes were taken into account for the methodology variable: Tactical Game in Basketball (TGB), Direct Instruction in Basketball (DIB) and Service Teacher´s Basketball Unit (STBU). TGB was based on the Tactical Game Approach (TGA) and DIB on the Direct Instruction (DI) method [39]. Both programmes had been previously validated [40] and were equivalent for each methodology in the number of tasks, contents and game phases (*p* > 0.05). The programmes were valid and reliable for teaching basketball in the school context, as they surpassed the critical point (*V* > 0.70) and obtained excellent internal validity with a value of (α = 0.96). An external teacher, who was also a researcher in the field of Sports Pedagogy and a basketball coach, implemented these programmes. The STBU programme was designed and implemented by an in-service teacher with 28 years’ experience as a primary school teacher. This teacher had total freedom in the design of the teaching unit on basketball.

Figure 2 shows the descriptive analysis of each of the intervention programmes (DI, TGA and STBU), bearing in mind some of the variables that influence task intensity: game situation, teaching means and playing space. The definition and categorisation of the variables was performed using the SIATE [25], adapting the initial proposal in order to reduce the number of categories in the game situation and teaching means categories. The tasks were categorised by an external assessor with specific training in the research topic.

According to the use of the different variables by the in-service teacher, it can be seen that the STBU programme was very similar to the DI method, predominantly with tasks without opposition, developed as static activities or performed in large spaces, highlighting exercises as the teaching–learning means. Bearing in mind the scientific literature, these characteristics are close to teaching models based on direct instruction [11,41,42].

The study dependent variables were the physical and physiological demands recorded by the students during the intervention. Specifically, a total of 30 objective dependent variables were recorded, identified as external (eTL) and internal (iTL) load variables.

The eTL variables used were (i) distance in metres (dis_m); (ii) metres covered per minute (m/min); (iii) number of accelerations (Nacc); (iv) accelerations per minute (acc/min); (v) number of decelerations (Ndec); (vi) decelerations per minute (dec/min); (vii) maximum speed (MAX Speed); (viii) average speed (AVG Speed); (ix) percentage of time devoted to high-intensity activities (HIA% ≥ 12.7 Km/h); (x) percentage of time devoted to walking (% walking ≤ 5.2 km/h); (xi) percentage of time devoted to jogging (% jog. = 5.2–10.5 km/h); (xii) percentage of time devoted to running (% running = 10.5–15.7 km/h); (xiii) percentage of time devoted to sprinting (% sprinting ≥ 15.7 km/h); (xiv) number of sprints (Nsprints); (xv) number of impacts received (Nimpacts); (xvi) number of steps (Nsteps); (xvii) steps per minute (steps/min); (xviii) number of jumps (Njumps); (xix) jumps per minute (jumps/min); (xx) total player load (PL); and (xxi) total player load per minute (PL/min). The variables acc/min and dec/min quantify the speed changes (m/s^2^) performed during the session/match and indicate both positive and negative changes. These variables are associated with specific movements in basketball, like starts, stops and changes of direction [43].
(1)$$\bar{a}=\frac{v-v_{0}}{t}$$


The PL variable is calculated from the accelerations using the following formula [44]:
(2)$$PL_{n}=\frac{\sqrt{{\left(x_{n}-x_{n-1}\right)}^{2}+{\left(y_{n}-y_{n-1}\right)}^{2}+{\left(z_{n}-z_{n-1}\right)}^{2}}}{100}$$


The iTL variables used in the study were (i) maximum HR (HRmax); (ii) average HR (AVG HR); (iii) percentage of relative HR (rel HR%); (iv) percentage of time spent in the HR range of 50 to 60% (50–60% HR); (v) percentage of time spent in the HR range of 60 to 70% (60–70% HR); (vi) percentage of time spent in the HR range of 70 to 80% (70–80% HR); (vii) percentage of time spent in the HR range of 80 to 90% (80–90% HR); (viii) percentage of time spent in the HR range of 90 to 95% (90–95% HR); and (ix) percentage of time spent in the HR range of 95 to 200% (95–200% HR). The formula proposed by Whaley et al. [45] was used to calculate maximum HR:

Men
(3)
HR*max* (*bpm*) = 203.9 − [0.812 × age] + [0.276 × HR*basal*] − [0.084 × *Body Mass*]



Women
(4)
HR*max* (*bpm*) = 204.8 − [0.718 × age] + [0.162 × HR*basal*] − [0.105 × *Body Mass*]



Lastly, the ratings of perceived exertion (RPE) (subjective iTL) recorded by the students in the basketball practices during the assessment tests (pre-test and post-test) were analysed.

### 2.4. Instruments

For the eTL data collection variables, each student was equipped with a WIMU Pro^TM^ (Real Track Systems, Almería, Spain) inertial device, which consisted of an inertial recording system that made it possible to record and monitor physical activity and movement in real time. To record the iTL variables, each instrument was synchronised with a GARMIN^TM^ heart rate band. Lastly, SPRO^TM^ software (Real Track Systems, Almería, Spain) was used to convert the information collected into quantitative data.

The students’ RPE during the assessment tests (pre-test/post-test) was recorded using the Borg scale [28], adapted for children by Eston and Parfitt [46]. In this new version, the degree of perceived effort is represented with pictograms and a quantitative scale from 0 to 10, with 0 being the minimal degree of effort and 10 the maximum. Lastly, the data collected with the SPRO^TM^ software and the RPE scale were exported to the SPSS 24 (IBM Corp. 2016. IBM SPSS Statistics para Windows, Version 24, IBM Corp, Armonk, NY, USA) statistical programme for processing.

### 2.5. Procedure

#### 2.5.1. Prior Planning

To implement the intervention in a school, it was necessary to obtain several authorisations. First, approval of the study was sought from the University Bioethics Committee (Ref. 247/2019). Then, at the beginning of the school year, the authorisation of the management team and the Physical Education teacher was requested to be able to carry out the study in the school. Once authorisation had been obtained from the school management and Physical Education teacher, a key figure in the study, informed consent was requested from the parents or legal guardians of the students, following the ethical guidelines of the Declaration of Helsinki and Organic Law 15/1999 of 13 December on the protection of personal information (LOPD) (BOE 14 December 1999). After all the relevant authorisations had been obtained, there was a meeting with the Physical Education teacher to plan the duration of the intervention and be able to adapt the teaching programme to the study requirements. It was finally decided to carry out the intervention during the third trimester, specifically during April, May and June of 2019.

#### 2.5.2. Intervention

First, an initial practical assessment was performed (pre-test), in which a total of ten 3 vs. 3 matches were played, five on each half of the basketball court. Each match lasted six minutes, and students had two minutes between each match to rest and record their RPE. The teams were organised beforehand by the teacher-researcher. In order to form homogeneous teams, the level of declarative and procedural knowledge of basketball was used as a reference, obtained with the instrument Declarative and Procedural Knowledge in Basketball (TDPKB) [47]. Then the application of the intervention programmes began. Each programme comprised nine sessions of one hour each. In each session, an audio recording was made of the teacher’s communications in order to review the procedure and ensure the adaptation of the intervention to the previously planned teaching methodologies, and in the case of the in-service teacher, to analyse the teaching intervention process.

Lastly, once the intervention programmes on basketball had been implemented, a final practical assessment (post-test) was conducted, where the students again played 3 vs. 3 matches on half the court, following the same planning and structure as the initial assessment. The practical intervention lasted 11 h, counting sessions and practical assessments. The volume of practice was adequate, as interventions of more than eight hours’ duration are associated with better results [32]. At the beginning of each practical intervention (sessions and assessments), each student was monitored with a WIMU Pro^TM^ inertial device and a GARMIN^TM^ heart rate band. This monitoring process was made even more efficient with the participation of a second researcher, so that approximately ten minutes were necessary in each intervention to place the devices.

#### 2.5.3. Data Processing

The data recorded by the WIMU Pro^TM^ inertial devices and the GARMIN^TM^ heart rate bands were converted into quantitative data using SPRO^TM^ software. Subsequently, a cluster analysis was performed to establish the HIA values and the speed ranges (walking, jogging, running and sprinting), adapting them to the students’ characteristics [37]. The SPRO^TM^ software was adapted according to the ranges obtained from the cluster analysis. This procedure was carried out because the software fixes sprints at 21 km/h by default, a difficult speed for students of primary education to attain given their physical characteristics. Finally speeds of 15.6 km/h or over were considered sprints, and the HIA value was considered as 12.7 km/h.

### 2.6. Statistical Analysis

First the Kolmogorov-Smirnov, Levene and Rachas [48] tests were used for the assumption of criteria in order to establish the adequate model for the confirmation of the hypotheses. The results showed that the study variables did not comply with the assumption of normality, homogeneity and randomness, so that non-parametric mathematical models were used to confirm the hypotheses.

A descriptive analysis was made of the variables according to the method and prior experience during the implementation of the programme (sessions) and during the practical assessments (3 vs. 3 matches). The Kruskal-Wallis H test for independent samples was used for the inferential analysis to determine the effect of the method, and the Mann-Whitney U test was used to determine the effect of experience [48]. The inferential analysis, according to experience, was only performed with the pre-test and post-test, as the three groups were exposed to the same workloads by practising the same tasks (3 vs. 3 matches). The assumption of criteria tests were subsequently performed to analyse the RPE recorded in the assessment tests, according to the method and prior experience. It was determined that parametric tests should be used for the assessment (pre-test): a t-test for independent samples and a one-way ANOVA. For the post-test assessment the Kruskal-Wallis H and Mann-Whitney U non-parametric tests were used. Lastly, Wilcoxon’s T test for related samples was used to analyse the evolution of each group between the pre-test and post-test according to the method and experience. The degree of association between the objective and subjective ITL of the students was analysed. For this, the Pearson and Spearman correlation test was used, interpreted according to [48] as follows: insignificant (r^2^ < 0.1), small (0.1 < r^2^ < 0.3), moderate (0.3 < r^2^ < 0.5), large (0.5 < r^2^ < 0.7), very large (0.7 < r^2^ < 0.9), almost perfect (r^2^ > 0.9) and perfect (r^2^ = 1).

Finally, the effect size of the statistical analyses of the non-parametric tests was calculated using epsilon squared (E^2^_R_) and Rosenthal’s r (r) [49]. Cohen’s d and partial eta squared (η^2^) were used for the parametric tests [50,51]. Cohen’s d effect sizes were considered as small (0.200–0.499), medium (0.500–0.799) and large (>0.800) [50]. With regard to (η^2^), the range was small (0.010–0.059), medium (0.060–0.139) and large (>0.140) [52]. For (E^2^_R_) the range was small (0.01–0.08), medium (0.08–0.26) and large (≥0.26) [49]. Lastly, for Rosenthal’s r, the criteria were the same as those described for Cohen [50].

## 3. Results

Table 1 and Table 2 show the results of the differences in eTL and iTL among the intervention programmes during the application of the sessions and the assessment tests, according to the teaching methodology used: DI, TGA and STBU.

The results show significant differences among the three groups in all the variables of eTL and iTL, except in HR in the range of 70–80%. Pairwise comparisons show that the programme based on the TGA method obtained better results in all the variables of eTL except (acc/min), (dec/min), (jumps/min) and in HIA, walking. With respect to the iTL values, the TGA programme obtained better results than the DI and STBU in the variables HR/max (DI = 164.46, TGA = 176.98, STBU = 162.46), *HR*/*AVG* (DI = 142.94, TGA = 157.12, STBU = 143.98) and Rel/HR% (DI = 72.82, TGA = 82.56, STBU = 78.50). Furthermore, these students worked for longer in the high-intensity HR ranges 90–95% HR (DI = 6.50, TGA = 17.07, STBU = 10.95) and 95–200% HR (DI = 3.45, TGA = 9.69, STBU = 7.27) and with greater workload PL (DI = 4.92, TGA = 6.95, STBU = 2.99) and PL/min (DI = 0.66, TGA = 0.93, STBU = 0.76). Finally, the results on the size of the effect show that the differences are large (E^2^_R_ ≥ 0.26) in the variables dis/m, Nacc, Ndec, Nsteps, Nimpacts and PL.

The results show significant differences in all the eTL variables, except in the variables sprints%, Nsprints, Nsteps, Njumps and Nimpacts. The trend in the results is similar to that obtained in the sessions: the TGA method obtained higher levels of eTL, except in the acc/dec. Regarding iTL, significant differences were only obtained in the heart rate range 90–95% HR between method STBU and DI (DI = 21.48, TGA = 25.33, STBU = 27.44). The TGA method did not obtain higher levels of HR. However, these students covered greater distance (DI = 260.44, TGA = 354.81, STBU = 346.03) and spent most of their time doing high-intensity activities, in the speed ranges of running (DI = 3.42, TGA = 11.26, STBU = 8.32) and sprinting (DI = 0.00, TGA = 0.12, STBU = 0.11). The results on the size of the effect show that the differences are medium (E^2^_R_ ≥ 0.26) in the variables dis/m, Nacc, Ndec, Nsteps, Nimpacts and PL.

Table 3 below shows the results of the differences in iTL and eTL according to the prior experience of the students during the assessment tests.

The results obtained during the evaluations indicate that there are no differences in most variables with respect to experience. The variables with differences were Max_Speed (Experience = 11.50, No experience = 11.77), HR_max (Experience = 185.40, No experience = 179.57) AVG_HR (Experience = 166.97, No experience = 160.88) and Rel_HR (Experience = 84.09, No experience = 81.64). The PL achieved by students was very similar (Experience = 6.61, No experience = 6.62); e however, experienced students showed higher HR and worked for longer in the high-intensity ranges of 90–95% HR (Experience = 26.46, No experience = 23.33) and 95–200% HR (Experience = 23.70, No experience = 19.70).

The results of the differences in subjective iTL (RPE) recorded in the assessment tests according to experience and teaching methodology are shown in Table 4 and Table 5, respectively.

Wilcoxon’s t test for related samples found no differences among the assessment tests according to teaching method (DI, *p* = 0.05; TGA, *p* = 0.51; STBU, *p* = 0.93). There were also no differences regarding students’ prior experience among the assessment tests of students who had not practised basketball before (*p* = 0.87). However, significant differences were found for the students who did have experience (*p* = 0.04*; r = 0.47), as they indicated lower levels of perceived exertion in the post-test assessment.

Table 6 below shows the correlational analysis between subjective and objective iTL.

The results show that there is no correlation between the objective and subjective iTL (*p* > 0.05); therefore, the correlation strength is weak in all the variables (r < 0.25).

## 4. Discussion

According to the scientific literature, the methodology used by the Physical Education teachers in their classes conditions the students’ physical aptitude, as it implies different workloads [15]. Thus, the objective of the present study was to quantify and compare eTL, iTL and RPE according to the teaching methodology and the students’ prior experience, after the application of three different intervention programmes in school basketball, one of which was designed and implemented by an in-service teacher. The results identified significant differences in favour of the students who received the TGB programme, based on the TGA teaching method. During the sessions, they recorded higher values of eTL (player load; DI = 4.92, TGA = 6.95, STBU = 2.99) and iTL (mean heart rate; DI = 142.94, TGA = 157.12, STBU = 143.98). During the evaluation tests, they presented heart rate levels similar to those obtained by the students in the rest of the programmes. However, they spent more time doing high-intensity activity, working longer in the running (DI= 3.42, TGA= 11.26, STBU= 8.32) and sprinting speed ranges (DI = 0.00, TGA = 0.12, STBU = 0.11). Regarding the RPE recorded in the evaluations, the results show that experience and the intervention programmes had no effects (*p* > 0.05) on the students’ RPE. However, there was an evolution in the three class groups, registering a more efficient RPE at the end of the intervention (post-test).

During the sessions, the results showed significant differences among the three groups in all the variables of eTL and iTL, except in HR in the range of 70–80%. Therefore, the design of the tasks according to different approaches directly influences the workload that the students support. Pairwise comparisons show that the programme based on the TGA method obtained better results in all the variables except (acc/min), (dec/min), (jumps/min) and in HIA, walking. This could be due to the characteristics of the tasks in the rest of the programmes, specifically in the execution of movements and their duration [53]. The organisation of the tasks based on the DI method implies that the students are arranged in rows, waiting for their classmates to perform their short practical intervention. Thus, the task begins at a walking pace but ends quickly at greater intensity, while in the TGA method the interventions develop continuously, beginning with low-intensity acceleration and gradually increasing to high speeds [54], explaining the differences in the (acc/min) and (dec/min) variables. Regarding the number of jumps per minute, it was lower in TGA because the tasks based on traditional methodology focus on repeating the same action always following the same movement pattern, as occurs, for example, in throw or rebound actions [6]. With respect to the iTL values, the TGA programme obtained better results than the DI and STBU in the (HR/max) and (HR/AVG) variables. Furthermore, these students worked for longer in the high-intensity HR ranges. These results indicate that the tasks using the TGA method were performed at greater intensity, which is confirmed in the analysis of the (PL) and (PL/min) variables [55]. After analysing the students’ physiological demands, it can be seen that the loads supported by the students using the TGA method are very similar to those recorded in actual play [56]. This is because of the design of the tasks, as all of them involve opposition and mostly take place in smaller areas, variables that increase the complexity and intensity of the task. Regarding the STBU programme, the descriptive analysis shows that the characteristics of the pedagogical variables are very similar to DI. Both focus mainly on tasks without opposition, using exercises and characteristics representative of a traditional methodology [11]. These similarities are reflected in the students’ physical and physiological responses, as both methods obtained results that are comparable in PL and HR (PL/min; DI = 0.66 and STBU = 0.75) (HR/AVG; DI = 142.94 and STBU = 143.98); however, in the STBU programme, less distance was covered, and the students worked in HR intervals of greater intensity.

Similar results were obtained in the study by González-Espinosa et al. [6]: the TGA method was the one that recorded the highest iTL and eTL values. However, there are differences in the study by García-Ceberino et al. [37] on school soccer. The students who participated in the TGA method also recorded higher values in iTL; however, the students following the DI method were the ones who recorded higher values of eTL, which could be due to the characteristics of the sample and the tasks designed. Of the DI group, 42.9% practised soccer as an out-of-school activity during the application of the programmes, while in the TGA group only 15% of the participants practised outside of school. Students have different levels of knowledge according to their experience in sports practice [57]. More knowledge about the sport allows greater interaction in the game, obtaining higher levels of eTL. Regarding the design of the tasks, they were predominantly without opposition, organised in rows, and provoked greater accelerations and decelerations than the TGA method. These movements are closely related to the workload [58]. However, independent of the experience and the movements performed, the TGA method is suitable for enhancing the physical fitness and health of students [6,37]. For this reason, it is appropriate to use teaching models based on tactics for planning Physical Education sessions.

During the assessment tests (pre-test and post-test) the students performed the same tasks (3 vs. 3 matches). However, the results showed significant differences, although in fewer variables compared to the sessions. The tendency in the results was similar, the TGA method obtained higher levels of eTL, except in the acc/dec variables, as the students who had followed the traditional designs continued to experience greater changes of pace, making more movements but perhaps with less efficacy. In this case it could be due to the play level of the students, as the more experienced players recorded fewer accelerations [59] due to their knowledge and control of the game. The players with a greater knowledge of the game make better use of their movements, which are thus more effective. Regarding iTL in the assessments, the TGA method did not obtain higher HR levels. However, these students covered a greater distance and spent most of the time performing high-intensity activities in the speed ranges of running and sprinting. Therefore, they showed greater physical fitness due to their prior training. González-Espinosa et al. [6] state that higher intensity training sessions lead to better attention, perception, anticipation or planning, and consequently, responses are developed more quickly and efficiently. The pairwise comparisons showed differences between the TGA and STBU method and DI, with the latter being the one that provoked the lowest physical and physiological demands during the assessment. The students who followed the DI method covered less distance and at a slow speed, did not sprint during the match and spent most of the time in the speed range of walking; in addition, their high-intensity activity was noticeably lower than the rest of the programmes. This could be due to the characteristics of their previous training, specifically, the design of the tasks used during the sessions. In the DI programme, the organized tasks in rows and with not much interaction between individuals are used more frequently, thus providing a low intensity to that activity.

Bearing in mind prior experience, 52.6% of the students who participated in the TGA method practised basketball as an out-of-school activity. In the DI group, the percentage was 44.4%, and in the STBU programme designed by the teacher, 27.8% of the students practised basketball. The improvements obtained with the TGA method during the evaluation could be due to prior experience, as this was the group with the highest number of experienced students. However, the results obtained during the assessments indicate that there were no differences in the majority of the variables regarding experience. In the variables obtained, differences were not favourable for the TGA (Max/Speed, HR/max, AVG/HR and Rel/HR). Thus, the improvements in the physical and physiological demands were not conditioned by the experience variable, but rather by the teaching methodology. The PL attained by the students was very similar; however, the experienced students recorded higher HR, which could be due to their working for longer in the high-intensity ranges. These results do not follow the tendency of results from sports training. Torres-Ronda et al. [60] state that more experienced players show a higher eTL associated with a lower iTL, due to differences in physical aptitude. This discrepancy could be because of the sociodemographic characteristics of the sample, for example, the time devoted to basketball as an out-of-school activity, if it is carried out with competitive aims or is at the initiation stage, or to the methodology used by the coach during the season.

Lastly, regarding the subjective iTL (*RPE*) recorded in the assessment tests, the results show that experience and the intervention programmes had no effects (*p* > 0.05) on the students’ RPE. However, the means indicate that there was evolution in the RPE between the pre-test/post-test in the TGA and DI methods, with more efficient values being recorded in the post-test. The STBU programme provoked a higher RPE during the post-test, which could be due to the large difference in the distance covered by the students between the application in the sessions (dis/m = 156.50) and the assessment tests (dis/m = 346.03). The differences due to experience showed that the experienced students recorded higher values of RPE in the pre-test and lower values in the post-test; there was an evolution in the RPE of both groups between the assessment tests. Similar results were obtained by García-Ceberino et al. [37], as the methodology used had no effect on RPE, and more efficient values were recorded in the post-test. Thus, sports practice, independently of the method used, improves RPE in the student, becoming more efficient once the training period comes to an end. Sperlich et al. [4] showed that both duration and the intensity with which a physical activity is practised are primordial factors for obtaining health benefits. In this study, the distance covered, movement intensity, PL and HR ranges were greater when the TGA methodology was used, both in the sessions and in the assessment tests. Therefore, this method permits a greater development of physical aptitude, increasing aerobic capacity and thus the health of the students [17]. Other studies also found differences in student learning when using pedagogical approaches different to the traditional one. However, in order to obtain these benefits, continuing teacher training is important, as they must learn to teach through these approaches. Teachers need support to make this conceptual change in teaching; for this, a collaborative role between university and school is necessary to create learning communities that focus on the current needs of education [20,21].

## 5. Conclusions

The demands in eTL and iTL produced during the sessions using the TGA method were greater than those provoked by the rest of the programmes: DI and STBU. Furthermore, they were more similar to those produced by real play (3 vs. 3 matches).

The improvements in the physical and physiological demands produced by the TGA method during the assessments were not conditioned by the experience variable, which identified few significant differences; however, there was a close relation with the teaching methodology used. Therefore, the importance of teachers’ task planning and design is confirmed, as adequate planning makes it possible to attain the proposed objectives and satisfy the physiological needs of the students, optimising their physical aptitude and enhancing the increase in physical and physiological demands during class practice. For this reason, methodologies centred on the student, like the TGA method, are recommended.

### 5.1. Practical Application

This study offers Physical Education teachers relevant information on the effects produced by different teaching methods on the students’ physical and physiological demands. Given the results obtained, the TGA method is recommended for teaching basketball in the school context, as it achieves a greater eTL and iTL. Thus, teachers should bear this method in mind in their daily planning to obtain the total development of the student.

The data are conclusive and important, both for the educational context and sports initiation, as they offer guidance to teachers and coaches on the benefits of this method for training students and novice players.

### 5.2. Strengths and Limitations

Only two studies [6,37] have investigated the workload supported by primary school students after the application of intervention programmes based on different teaching methods. This study complements this line of research, as it uses the planning and intervention of an in-service Physical Education teacher, a fundamental figure in the teaching–learning process. It also makes it possible to see the teacher’s position regarding the teaching methodology for invasion sports.

Among the limitations to this study, it should be underlined that the results obtained were from a small sample with specific characteristics. Therefore, it is necessary to continue to investigate, expanding the sample size and varying the profile of the subjects with the aim of achieving greater experimental control. It would also be necessary to increase the number of programmes designed and imparted by in-service teachers, as the collaboration of the teacher is imperative to be able to continue to encourage research in schools.

## Figures and Tables

**Figure 1 ijerph-18-11854-f001:**
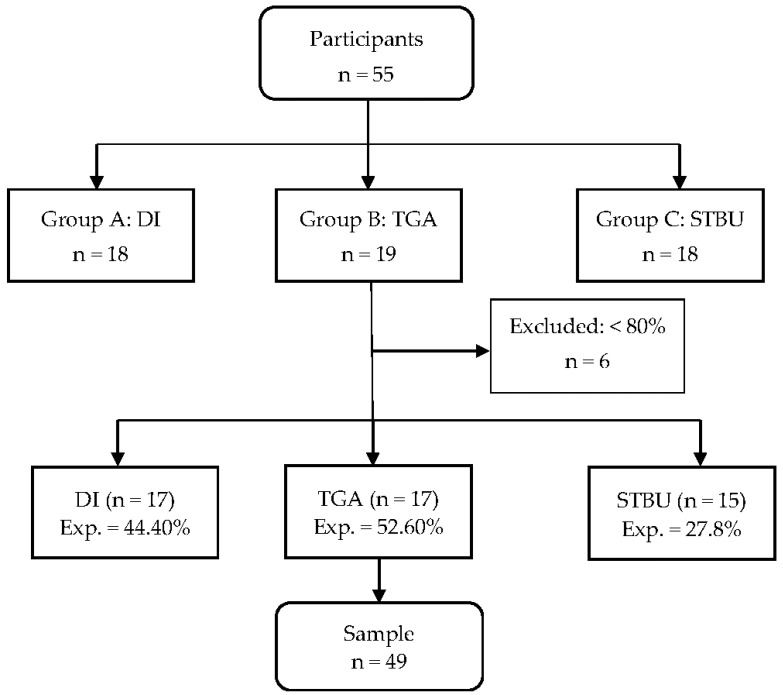
Flow chart. Note: Exp., previous experience in basketball; n, number.

**Figure 2 ijerph-18-11854-f002:**
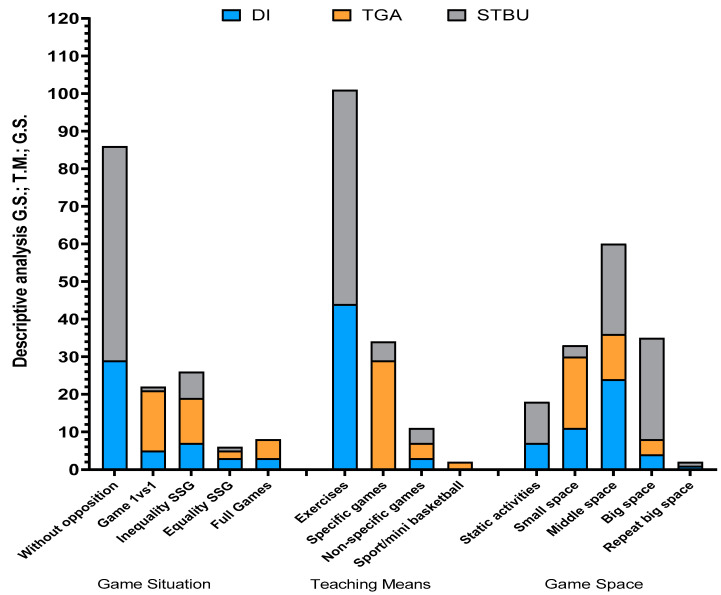
Descriptive analysis of intervention programs according to pedagogical variables. Note: DI, Direct instruction; TGA, Tactical Game Approach; STBU, Service Teacher´s Basketball Unit; SSG, Small-Sided Games.

**Table 1 ijerph-18-11854-t001:** Results of the application of the programmes according to the teaching methodology.

Study Variables		DI M ± SD	TGA M ± SD	STBU M ± SD	*H*	*p*	E^2^_R_	Pairwise Comparisons
Kinematic eTL	dis(m)	283.15 ± 150.90	373.94 ± 147.06	156.50 ± 88.46	821.02	0.00 *	0.37	TGA>DI*/TGA>STBU*/DI>STBU*
	m/min	37.54 ± 18.05	50.07 ± 18.60	40.54 ± 23.25	118.45	0.00 *	0.05	TGA>DI*/TGA>STBU*
	Nacc	276.27 ± 79.61	280.72 ± 122.32	172.16 ± 111.30	672.59	0.00 *	0.30	TGA>STBU*/DI>STBU*
	acc/min	37.99 ± 5.84	35.33 ± 4.11	38.01 ± 5.87	104.25	0.00 *	0.05	DI>TGA*/STBU>TGA*
	Ndec	275.93 ± 79.50	280.61 ± 122.24	172.08 ± 111.37	671.24	0.00 *	0.30	TGA>STBU*/DI>STBU*
	dec/min	37.94 ± 5.84	35.32 ± 4.10	37.98 ± 5.86	101.70	0.00 *	0.05	DI>TGA*/STBU>TGA*
	MAX Speed	10.28 ± 3.40	12.50 ± 2.64	9.36 ± 3.54	308.54	0.00 *	0.14	TGA>DI*/TGA>STBU*/DI>STBU*
	AVG Speed	3.46 ± 1.11	3.78 ± 0.80	3.37 ± 1.29	59.51	0.00 *	0.03	TGA>DI*/TGA>STBU*
	HIA %	4.21 ± 7.32	7.53 ± 8.07	5.06 ± 9.61	176.98	0.00 *	0.08	TGA>DI*/TGA>STBU*/DI>STBU*
	walk %	71.96 ± 21.54	66.66 ± 16.01	71.52 ± 25.50	39.91	0.00 *	0.02	DI>TGA*/STBU>TGA*
	jog %	25.02 ± 19.17	27.22 ± 12.52	24.61 ± 21.37	19.72	0.00 *	0.01	TGA>DI*/TGA>STBU*
	run %	2.99 ± 5.00	5.96 ± 6.63	3.74 ± 7.43	193.38	0.00 *	0.09	TGA>DI*/TGA>STBU*/DI>STBU*
	sprint %	0.03 ± 0.35	0.16 ± 0.84	0.12 ± 1.05	43.86	0.00 *	0.02	TGA>DI*/TGA>STBU*
	Nsprints	0.01 ± 0.11	0.11 ± 0.55	0.04 ± 0.34	28.67	0.00 *	0.01	TGA>DI*/TGA>STBU*
	Nsteps	278.56 ± 181.93	396.64 ± 176.11	165.34 ± 113.59	650.85	0.00 *	0.29	TGA>DI*/TGA>STBU*/DI>STBU*
	steps/min	36.81 ± 22.26	53.92 ± 25.08	43.23 ± 31.58	118.11	0.00 *	0.05	TGA>DI*/TGA>STBU*
	Njumps	2.93 ± 3.42	3.61 ± 3.76	1.49 ± 2.47	224.01	0.00 *	0.10	TGA>DI*/TGA>STBU*/DI>STBU*
	jumps/min	0.57 ± 0.42	0.57 ± 0.43	0.73 ± 0.61	16.81	0.00 *	0.01	STBU>TGA*/STBU>DI*
Neuromuscular eTL	Nimpacts	904.54 ± 499.55	1273.29 ± 458.24	497.31 ± 285.63	903.01	0.00 *	0.41	TGA>DI*/TGA>STBU*/DI>STBU*
	PL	4.92 ± 2.46	6.95 ± 2.57	2.99 ± 1.65	868.09	0.00 *	0.39	TGA>DI*/TGA>STBU*/DI>STBU*
	PL/min	0.66 ± 0.28	0.93 ± 0.33	0.76 ± 0.42	184.52	0.00 *	0.08	TGA>DI*/TGA>STBU*/STBU>DI*
iTL	HRmax	164.46 ± 26.32	176.98 ± 18.35	162.46 ± 22.72	166.10	0.00 *	0.08	TGA>DI*/TGA>STBU*/DI>STBU*
	AVG HR	142.94 ± 24.21	157.12 ± 18.57	143.98 ± 23.42	160.12	0.00 *	0.07	TGA>DI*/TGA>STBU*
	Rel HR %	72.82 ± 12.27	82.56 ± 8.19	78.50 ± 10.19	252.21	0.00 *	0.12	TGA>DI*/TGA>STBU*/STBU>DI*
	50–60% HR	11.61 ± 20.99	2.37 ± 7.15	6.66 ± 17.15	132.08	0.00 *	0.06	DI>TGA*/DI>STBU*
	60–70% HR	20.75 ± 24.22	8.94 ± 14.62	15.29 ± 21.89	84.52	0.00 *	0.04	DI>TGA*/DI>STBU*/STBU>TGA*
	70–80% HR	28.42 ± 25.71	23.76 ± 21.39	27.54 ± 25.55	4.56	0.11	0.00	-
	80–90% HR	21.90 ± 24.21	36.83 ± 23.70	30.42 ± 26.88	134.00	0.00 *	0.06	TGA>DI*/TGA>STBU*/STBU>DI*
	90–95% HR	6.50 ± 13.34	17.07 ± 18.42	10.95 ± 16.20	177.59	0.00 *	0.08	TGA>DI*/TGA>STBU*/STBU>DI*
	95–200% HR	3.45 ± 12.38	9.69 ± 16.69	7.27 ± 15.58	126.38	0.00 *	0.06	TGA>DI*/TGA>STBU*/STBU>DI*

Note: M, mean; SD, standard deviation; H, Kruskal-Wallis H; E^2^_R_, epsilon squared; TGA, Tactical Games Approach; DI, Direct Instruction; STBU, Service Teacher´s Basketball Unit; N, number; min, minute; HR, heart rate, dis (m), distance in metres; m/min, metres covered per minute; Nacc, number of accelerations; acc/min, accelerations per minute; Ndec, number of decelerations; dec/min, decelerations per minute; MAX Speed, maximum speed; AVG Speed, average speed; HIA%, percentage of time devoted to high-intensity activities; PL, total player load; HRmax, maximum heart rate; AVG HR, average heart rate; Rel HR %, percentage of relative heart rate; HR 50–60%, heart rate range of 50–60%; * *p* < 0.05; Pairwise Comparison*, there are significant differences.

**Table 2 ijerph-18-11854-t002:** Results of the assessment tests (pre-test/post-test) according to the teaching methodology.

Study Variables		DI M ± SD	TGA M ± SD	STBU M ± SD	*H*	*p*	E^2^_R_	Pairwise Comparisons
Kinematic eTL	dis(m)	260.44 ± 105.90	354.81 ± 101.61	346.03 ± 141.08	56.82	0.00 *	0.14	TGA>DI*/STBU>DI*
	m/min	42.37 ± 17.02	64.08 ± 11.96	57.54 ± 23.08	95.14	0.00 *	0.23	TGA>DI*/STBU>DI*
	Nacc	219.01 ± 36.38	175.31 ± 23.03	208.24 ± 48.89	93.95	0.00 *	0.23	DI>TGA*/DI>STBU*/STBU>TGA*
	acc/min	35.95 ± 5.91	32.63 ± 5.13	34.52 ± 7.88	30.61	0.00 *	0.07	DI>TGA*/DI>STBU*
	Ndec	219.04 ± 36.55	175.21 ± 23.24	208.22 ± 49.03	94.09	0.00 *	0.23	DI>TGA*/DI>STBU*/STBU>TGA*
	dec/min	35.95 ± 5.95	32.60 ± 5.10	34.52 ± 7.91	31.08	0.00 *	0.07	DI>TGA*/DI>STBU*
	MAX Speed	9.61 ± 3.23	13.17 ± 1.86	12.19 ± 3.45	85.27	0.00 *	0.20	TGA>DI*/STBU>DI*
	AVG Speed	3.20 ± 1.27	4.65 ± 1.48	4.22 ± 1.44	77.35	0.00 *	0.19	TGA>DI*/STBU>DI*
	HIA %	4.49 ± 12.01	14.59 ± 19.83	11.20 ± 17.54	89.49	0.00 *	0.22	TGA>DI*/STBU>DI*
	walk %	77.70 ± 25.31	52.89 ± 23.26	59.21 ± 23.68	79.32	0.00 *	0.19	DI>TGA*/DI>STBU*
	jog %	18.87 ± 19.89	35.73 ± 12.64	32.36 ± 16.14	73.03	0.00 *	0.18	TGA>DI*/STBU>DI*
	run %	3.42 ± 8.45	11.26 ± 15.85	8.32 ± 13.42	69.13	0.00 *	0.17	TGA>DI*/STBU>DI*
	sprint %	0.00 ± 0.04	0.12 ± 0.72	0.11 ± 0.66	5.63	0.06	0.01	-
	Nsprints	0.01 ± 0.08	0.06 ± 0.261	0.05 ± 0.22	5.12	0.08	0.01	-
	Nsteps	378.79 ± 190.59	425.03 ± 152.91	422.97 ± 222.80	4.53	0.10	0.01	-
	steps/min	60.75 ± 29.94	75.66 ± 18.70	70.28 ± 36.45	11.43	0.00 *	0.03	TGA>DI*/STBU>DI*
	Njumps	3.65 ± 3.62	4.46 ± 3.88	4.72 ± 4.71	4.25	0.12	0.01	
	jumps/min	0.75 ± 0.56	0.96 ± 0.55	0.96 ± 0.75	11.18	0.00 *	0.03	TGA>DI*/STBU>DI*
Neuromuscular eTL	Nimpacts	1126.04 ± 526.16	1148.39 ± 457.02	1056.44 ± 476.53	5.53	0.06	0.01	
	PL	6.09 ± 2.52	6.94 ± 2.06	6.83 ± 2.91	7.33	0.03 *	0.02	STBU>DI*
	PL/min	0.98 ± 0.39	1.25 ± 0.24	1.13 ± 0.48	27.26	0.00 *	0.07	TGA>DI*/STBU>DI*
iTL	HRmax	177.93 ± 33.60	182.03 ± 25.87	186.68 ± 22.65	2.46	0.29	0.01	-
	AVG HR	160.29 ± 35.78	161.82 ± 30.48	168.83 ± 26.73	2.65	0.27	0.01	-
	Rel HR %	80.58 ± 16.94	83.15 ± 12.63	84.52 ± 12.17	2.76	0.25	0.01	-
	50–60% HR	7.30 ± 18.03	6.76 ± 15.78	10.77 ± 24.50	0.31	0.86	0.00	-
	60–70% HR	6.63 ± 14.35	9.80 ± 17.90	8.98 ± 19.50	3.63	0.16	0.01	-
	70–80% HR	10.28 ± 17.75	8.96 ± 14.60	6.38 ± 9.82	3.13	0.21	0.01	-
	80–90% HR	24.23 ± 26.53	22.04 ± 22.95	21.93 ± 22.18	0.25	0.88	0.00	-
	90–95% HR	21.48 ± 22.07	25.33 ± 21.88	27.44 ± 21.43	6.71	0.04 *	0.02	STBU>DI*
	95–200% HR	19.60 ± 27.38	21.26 ± 24.56	23.64 ± 26.23	4.47	0.11	0.01	-

Note: M, mean; SD, standard deviation; H, Kruskal-Wallis H; E^2^_R_, epsilon squared; TGA, Tactical Games Approach; DI, Direct Instruction; STBU, Service Teacher´s Basketball Unit; N, number; min, minute; HR, heart rate, dis (m), distance in metres; m/min, metres covered per minute; Nacc, number of accelerations; acc/min, accelerations per minute; Ndec, number of decelerations; dec/min, decelerations per minute; MAX Speed, maximum speed; AVG Speed, average speed; HIA%, percentage of time devoted to high-intensity activities; PL, total player load; HRmax, maximum heart rate; AVG HR, average heart rate; Rel HR %, percentage of relative heart rate; HR 50–60%, heart rate range of 50–60%; * *p* < 0.05; Pairwise Comparison*, there are significant differences.

**Table 3 ijerph-18-11854-t003:** Results of the assessment tests (pre-test/post-test) according to prior experience.

Study Variables		Experience M ± SD	No Experience M ± SD	*U*	*p*	*r*
Kinematic eTL	dis(m)	314.96 ± 111.85	324.08 ± 133.70	20,437.00	0.41	0.04
	m/min	53.62 ± 17.55	55.42 ± 21.79	19,833.50	0.19	0.06
	Nacc	197.02 ± 38.76	203.69 ± 43.86	19,616.00	0.14	0.07
	acc/min	33.75 ± 6.09	34.86 ± 6.82	19,575.50	0.13	0.07
	Ndec	197.01 ± 38.88	203.64 ± 44.05	19,667.50	0.15	0.07
	dec/min	33.74 ± 6.08	34.85 ± 6.84	19,579.00	0.13	0.07
	MAX Speed	11.50 ± 3.00	11.77 ± 3.50	18,960.00	0.04 *	0.10
	AVG Speed	3.96 ± 1.44	4.06 ± 1.59	20,455.00	0.42	0.04
	HIA %	9.42 ± 17.11	10.60 ± 17.39	19,768.50	0.17	0.07
	walk %	64.63 ± 25.37	62.29 ± 26.99	19,787.50	0.18	0.07
	jog %	28.67 ± 17.71	29.16 ± 18.29	20,306.50	0.36	0.05
	run %	6.66 ± 12.13	8.45 ± 14.15	19,635.00	0.14	0.07
	sprint %	0.04 ± 0.41	0.11 ± 0.66	20,638.00	0.08	0.09
	Nsprints	0.02 ± 0.15	0.05 ± 0.24	20,888.00	0.16	0.07
	Nsteps	399.01 ± 177.92	416.40 ± 201.05	20,070.50	0.26	0.08
	steps/min	67.09 ± 27.23	70.28 ± 31.61	19,713.00	0.16	0.05
	Njumps	4.35 ± 4.12	4.21 ± 4.09	20,623.00	0.50	0.03
	jumps/min	0.86 ± 0.64	0.91 ± 0.63	12,678.50	0.37	0.90
Neuromuscular eTL	Nimpacts	1138.89 ± 474.94	1089.09 ± 497.95	19,469.50	0.11	0.08
	PL	6.61 ± 2.40	6.62 ± 2.65	21,194.50	0.84	0.01
	PL/min	1.12 ± 0.37	1.12 ± 0.42	21,273.00	0.89	0.01
iTL	HRmax	185.40 ± 25.59	179.57 ± 29.55	17,741.50	0.00 *	0.14
	AVG HR	166.97 ± 30.06	160.88 ± 32.28	18,165.00	0.01 *	0.13
	Rel HR %	84.09 ± 13.53	81.64 ± 14.59	18,758.00	0.04 *	0.10
	50–60% HR	6.63 ± 16.52	9.50 ± 21.87	19,689.00	0.09	0.08
	60–70% HR	7.55 ± 16.24	9.18 ± 18.19	18,952.00	0.04 *	0.10
	70–80% HR	7.40 ± 12.62	9.50 ± 15.87	20,385.50	0.44	0.04
	80–90% HR	22.78 ± 24.05	22.72 ± 23.93	20,956.00	0.77	0.01
	90–95% HR	26.46 ± 22.46	23.33 ± 21.37	19,610.00	0.16	0.07
	95–200% HR	23.70 ± 26.33	19.70 ± 25.78	19,555.50	0.14	0.07

Note: M, mean; SD, standard deviation; U, Mann-Whitney U; r, Rosenthal’s r; N, number; min, minute; HR, heart rate, dis (m), distance in metres; m/min, metres covered per minute; Nacc, number of accelerations; acc/min, accelerations per minute; Ndec, number of decelerations; dec/min, decelerations per minute; MAX Speed, maximum speed; AVG Speed, average speed; HIA%, percentage of time devoted to high intensity activities; PL, total player load; HRmax, maximum heart rate; AVG HR, average heart rate; Rel HR %, percentage of relative heart rate; HR 50–60%, heart rate range of 50–60%; * *p* < 0.05.

**Table 4 ijerph-18-11854-t004:** Results of the RPE scale in the assessment tests according to prior experience.

Experience	Yes (M ± SD)	No (M ± SD)	*t*/*u*	*df*	*p*	*dCohen*/*r*
Pre-test	4.27 ± 1.07	3.89 ± 1.38	1.01 ^a^	45	0.32	0.00
Post-test	3.51 ± 1.17	3.85 ± 1.48	259.50 ^b^	−0.50	0.62	0.07

Note: ^a^ T independent samples; ^b^ Mann-Whitney U; M, mean; SD, standard deviation; df, degrees of freedom; r, Rosenthal’s r.

**Table 5 ijerph-18-11854-t005:** Results of the RPE scale in the assessment tests according to methodology.

Methodology	DI (M ± SD)	TGA (M ± SD)	STBU (M ± SD)	*f*/*h*	*df*	*p*	η^2^/E^2^_R_
Pre-test	4.29 ± 1.29	3.96 ± 1.34	3.91 ± 1.18	1.32 ^a^	2	0.67	0.18
Post-test	3.49 ± 1.15	3.64 ± 0.93	4.00 ± 1.89	0.39 ^b^	2	0.83	0.01

Note: ^a^ Univariate linear model; ^b^ Kruskal-Wallis H; M, mean; SD, standard deviation; df, degrees of freedom; η^2^, partial eta squared; E^2^_R_, epsilon squared; TGA, Tactical Games Approach; DI, Direct Instruction; STBU, Service Teacher´s Basketball Unit.

**Table 6 ijerph-18-11854-t006:** Results of the correlation between subjective and objective iTL.

	M ± SD	r	*p*
RPE * HR_90–95%	3.88 ± 1.31/30.67 ± 14.33	0.01 ^a^	0.96
RPE * HR_95–200%	3.88 ± 1.31/27.73 ± 19.66	−0.05 ^a^	0.61
RPE * MAX_HR	3.88 ± 1.31/189.70 ± 18.11	0.10 ^b^	0.35
RPE * AVG_HR	3.88 ± 1.31/175.35 ± 19.60	0.07 ^b^	0.52

Note: M, mean; SD, standard deviation; r, correlation force; r ^a^, Pearson’s correlation; r ^b^, Spearman’s correlation; RPE, rated perceived exertion; HR 90–95%, heart rate range of 90 to 95%; HR 95–200%, heart rate range of 95–200%; MAX HR, maximum heart rate; AVG HR, average heart rate; * *p* < 0.05.

## Data Availability

Not applicable.

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
