# Peer review of "Influence of the Pedagogical Model and Experience on the Internal and External Task Load in School Basketball"

_ijerph, 2021, doi:10.3390/ijerph182211854_

Round 1

Reviewer 1 Report

First of all, thank you for the opportunity to review this very interesting experiment. In my opinion, there is a lack of such studies in sports pedagogy and psychology. Thus congratulations to the authors for such an idea and aim of the study. However, I have a few suggestions and comments I would like to share with the authors:

ABSTRACT

I would recommend adding that the investigated three teaching-learning programs were randomly assigned to the groups of children-students

INTRODUCTION

· In lines 37 to 39, I would cite "fresher" statistics for obesity, and we have 2021. 

· In lines 120 to 123, the authors should mention the second aim of the study, with the independent variable: the students' prior experience in basketball. The authors underline this aim at the beginning of the Discussion chapter, in line 338. Thus it should also be stated at the end of the Introduction chapter.

· I would generally recommend dividing the Introduction into more comprehensive paragraphs, such as the first paragraph: lines 34 till 60, the second paragraph: 61 till 90... It would give fluency to the Introduction.

MATERIAL AND METHODS

  • I would recommend the "Flow diagram of the participants. It usually makes the material about sample size more readable. However, this is merely a preference than a critical comment.
  • In line 182 should be eTL instead of iTL; these are external loads, not internal.
  • In line 212, the name of the subchapter is "Instruments and material." According to my knowledge, material means participants. So I would leave only the name "Instruments."
  • In lines 301 till 303, there is too small font size and should be corrected.

RESULTS

It would be easier for the reader to understand the results if the tables had the legends for all the abbreviations used in each table. Moreover, I would highly recommend describing each table carefully and its results. That would make the Results chapter more powerful and precise. 

DISCUSSION

In line 357, I would get rid of ellipsis and write all the pattern.

Lines 411 till 413- the sentence is hard to read and understand. Please correct it.

In line 423, the authors mention only higher HR as the only variable which improved after intervention among experienced students. However, there were more four significant differences between experienced and non-experienced students, which the authors should mention.

The same as Introduction I would recommend dividing the Discussion into broader paragraphs to make it more fluent.

In my opinion, the sample size is also a weakness. The small sample sizes and multiple analyses conducted attenuate confidence in the degree to which the focal results are statistically reliable. In my opinion, this is the most significant limitation. I kindly ask the authors to explain if I am wrong.

Author Response

"Please check the attachment"

Reviewer 2 Report

The manuscript entitled "Influence of the pedagogical model and experience on the internal and external task load in school basketball" presents an interesting topic for scientific and technical communities. However, major revision is necessary especially considering the manuscript conciseness. 

Abstract. Add numbers for better comprehension of the Results section; "Therefore, they presented better physical fitness." Merge this sentence to the previous sentence.

Introduction. Currently, society is continually changing. This sentence can be removed. . This situation has worsened since the negative effects of the current COVID- 19 pandemic have been analyzed [5]. Rewrite. . This section should be more concise. The authors can revise each paragraph and merge some of them to avoid unbalanced paragraph lengths.

Methods. Avoid the unbalanced paragraphs; "...were equivalent for each methodology in the number of tasks, contents, and game phases (p> 0.05). .(V> .70) and obtained excellent internal validity with a value of (α=.96). Space between numbers and symbols, abbreviations and symbols. . Note: DI, Direct instruction; TGA, Tactical game approach; STBU, Service teacher ́s basketball unit; SSG, Small-sided games. The authors can add the abbreviations in the table caption. . Present equations and formulas in the text following a correct order; This procedure was carried out because the software fixes 271 sprints at 21km/h. Give space between number and unit. . "...for the confirmation of the hypotheses." The hypotheses should be clearly described in the Introduction; . ..."(0.060–0.139) and large (> 0.140) [59]. For (E2R) the range was small (0.01-0.08), medium (0.08- 301 0.26) and large (≥0.26) [56]. Lastly for Rosenthal’s r the criteria were the same as those described for 302 Cohen [57]" Check the font size. 

Results . Table 1 to 6 captions: Provide a detailed Table caption. . Table 1 to 5 notes: Add to the caption and consider explaining all the abbreviations;

Discussion. The first paragraph should be rewritten and the hypotheses testing results should be stated;  following the same movement pattern (jump, throw...) [8]. Complete the examples inside the parentheses;

References: 67 references are too much. Reconsider the need of including them.

Author Response

"Please check the attachment"

Reviewer 3 Report

Thank you very much for the opportunity reviewing this manuscript. It is important to point out that physical education is the firts encounter students have at school in order to become physically active.

Now I would like to share my feedback as clear as possible. I would recommend to double check the language used in regards when mentioning methodology/approach/technique. This might be a problem for english as a first language readers, since these words aren´t synomyms. As such, I would recommend to change the language to approach, since the teaching approach used was the one compared during the study.

Introduction

I would recommend to cite the work of David Kirk (2010). Physical Education Futures when mentioning the teacher-centered approach used in PE, this would support your Metzler citation. I would also recommend to cite the work of Casey (2014). Models-Based practice, I think this work would  help to clarify the difference of using different approaches or models within the PE classes. As for a student-centered approach, I would recommend to cite the work of Oliver & Oesterreich (2013) since it may provide a different insight of what a student-centered looks like in PE.

Methodology

In the procedure section it was mentioned that the intervention was implemented in the months of may, june, etc... but does not mentions the year. I would recommend to add the year in order to clarify when the study collected data.

Discussion

Once the new references are added to the manuscript. I would suggest to add these citation as part of the discussion section. Especially, because as a way to support the findings and provide a clear idea of the different approaches implemented in the PE class.

Author Response

"Please check the attachment"
